# Experimental Study on Damage Characteristics of Copper-Reinforced Polytetrafluoroethylene Shaped-Charge Warhead Liner

**DOI:** 10.3390/polym14102068

**Published:** 2022-05-19

**Authors:** Jianya Yi, Zhijun Wang, Jianping Yin

**Affiliations:** College of Mechatronics Engineering, North University of China, Taiyuan 030051, China; wzj@nuc.edu.cn (Z.W.); yjp123@nuc.edu.cn (J.Y.)

**Keywords:** polymer composite, PTFE/Cu, mechanical strength, shaped-charge liner

## Abstract

Polymer materials have important applications in the4 terminal effect and damage by shaped-charge warheads. However, the low strength of pure PTFE materials reduces the penetrability of the expansive jet from these warheads, hindering its application. This study improves the strength of pure PTFE material by adding Cu powder to the shaped-charge liner. Three types of PTFE/Cu composites with different densities are prepared. The effect of increasing the density on the performance of an expansive jet is studied by a dynamic mechanical property experiment, microscopic analysis, numerical simulation, and a penetration experiment. The results show that the toughness and impact strength of the PTFE/Cu composites improve when 18–50.5% Cu is added. The strength of the composite increases linearly with the increase in Cu content. Numerical simulations and X-ray pulse experiments reveal that the addition of Cu powder enhances the cohesiveness of the head of the expansive jet. The jet head becomes more cohesive as the Cu content is increased. However, the length and diameter of the jet become smaller. The jet can create a deeper hole in the steel target and increase damage as more Cu is added to the liner.

## 1. Introduction

Polymer composites are widely used in defense infrastructure, navigation, aviation, and daily life because of their low density, low cost, and easy moldability. Poly-tetrafluoroethylene (PTFE), a thermoplastic polymer material, has a low friction coefficient, high thermal stability, and high chemical resistance. PTFE is suitable as a matrix material for polymer composites research owing to its high density. Several studies have investigated PTFE composites for defense infrastructure applications [1,2,3,4,5,6,7,8,9,10], which are mainly categorized into energetic materials and inert composites. Energetic materials are relatively insensitive under normal conditions (e.g., PTFE/Al, PTFE/Ti, and PTFE/Al/W), but undergo violent chemical reactions and release a large amount of chemical energy under high-strain-rate loading or high-pressure collision. Inert composites remain insensitive to external stressors (e.g., PTFE/Cu) [11,12,13,14,15,16].

This composite is primarily used in shaped-charge warheads owing to the ductility of Cu. Several researchers have studied PTFE/Cu [11,12,13]. For example, Wang [17] studied the electron contact transfer between PTFE and metal (Al/Cu). These research works have broadened our understanding of contact electron transfer between metals and nonmetals. In another research, Beckford [18] studied the effect of Cu nanoparticles on the friction properties of polydopamine (PDA)/PTFE films. The improvement in the wear resistance of films by nano-Cu particles was analyzed. Xie [19] studied the effects of the morphology and mass of Cu powder on the friction coefficient and wear properties of PTFE composites. Regarding the particle dispersion and interface bonding of composite materials, Wang [20] studied the successful preparation of aluminum matrix composites with high boron nitride nanosheets (BNNSs) content by variable speed ball milling, the gradual addition of BNNSs and finally direct current sintering (DCS). Such a ball milling method can effectively disperse BNNSs onto the existing Al surface and the fresh Al surface generated by ball milling, allowing for a high content of BNNSs homogeneously dispersed within the composites. Danaya [21] found that the morphology of polybutylene-adipate-co-terephthalate (PBAT)/thermoplastic star (TPS) showed aggregation of nanoparticles, resulting in poor mechanical properties. The interaction between ZnO nanofiller and the polymer increases the dispersion of nanoparticles and reduces the agglomeration of nanoparticles. Zhao [22] prepared reduced graphene oxide (RGO)/Cu Matrix composites by the electrostatic adsorption method with interface transition phase design. Adding Cu/Ti alloy powder can improve the bonding by forming carbides at the RGO/Cu interface, and finally improve the mechanical properties of the composites. Li [23] studied the preparation of composite solders with different graphene dispersion by different ball milling methods. Microstructure characterization showed that incoherent and amorphous interfaces were formed between graphene and tin. Guo [24] obtained a non-equilibrium interface that can provide tight interfacial bonding between the carbon nanotubes (CNTs) and Al matrix in the Al/CNTs composites fabricated through spark plasma sintering (SPS) and subsequently hot extrusion. This special interface, accompanied by small grain size, uniform dispersion, and the integrity of the CNTs, can significantly improve the mechanical properties of the Al/CNTs composite. In our last study [25], the mechanical properties of PTFE/Cu samples prepared by the hot pressing sintering process and extrusion process were discussed. The results show that the mechanical properties of the samples prepared by the hot pressing sintering process are better than those prepared by the extrusion process, but the penetration performance is weak as a liner.

Shape, particle size, composition, and processing technology are the key factors affecting the mechanical properties of composites. A study on PTFE shaped-charge liner aimed at achieving penetration without explosion found that pure PTFE had poor dynamic mechanical properties. Although it would not detonate the reactive armor, it had insufficient penetrability. Using Cu powder as the filler for filling modification can effectively improve the material density, and dynamic mechanical properties of pure PTFE under a high strain rate, as well as the penetrability of the jet [11,12,13,14,15,16]. Research on the dynamic mechanical properties of materials and structures under strong dynamic loads is germane to several engineering disciplines. In weapons technology, a comprehensive understanding of material deformation and damage in the process of high-speed collision helps design protective structures and ammunition. Therefore, the dynamic mechanical properties of the PTFE/Cu material must be studied and analyzed and the principle of its dynamic response under high strain rates must be fully understood, which is significant to warhead design.

In this study, three composites with different Cu contents were prepared by the extrusion process. The mechanical properties of these composites during impact were evaluated by the split-Hopkinson bar (SHPB) test. In addition, the influence of Cu content on the mechanical properties of the PTFE/Cu composites was analyzed. The formative properties of the expansive jet with different densities and its ability to damage steel targets were compared. These findings can be used to create a practical guide for the application of the composite as an ammunition material.

## 2. Experimental Materials

### 2.1. Preparation of Experimental Samples

DuPont 7A PTFE (220 μm) and oxygen-free high-conductivity Cu (3–5 μm) were used as substrates. The ball mill used was a planetary mill machine SN4.0 (Chao Yue, Shanghai, China), with a maximum volume of 4 L and operated at room temperature. The extruder used was a PFB150 (Shang Ke, Jiangsu, China), the working temperature was 260 °C, and the screw speed was 20–30 r/min. PTFE and Cu were crushed and screened, and subsequently added to the barrel. The raw materials were transferred to the barrel while compacting them by the rotation of the screw or the reciprocating push of the plunger. They were continuously melted in the barrel under the action of shear heat generation and an external heat source and pushed forward to the die under pressure. Finally, the PTFE/Cu rod was continuously extruded from the die.

If the mass fraction of PTFE and Cu powder is *C_PTFE_* and *C_Cu_*, respectively, and the volume is *V_PTFE_* and *V_Cu_*, respectively, there is:(1)$$V_{PTFE}=\frac{m_{Total}\cdot C_{PTFE}}{\rho_{PTFE}}$$
(2)$$V_{Cu}=\frac{m_{Total}\cdot C_{Cu}}{\rho_{Cu}}$$
(3)$$TMD=\frac{m_{Total}}{V_{PTFE}+V_{Cu}}$$

By substituting Formulas (1) and (2) into Formula (3), the Theoretical Maximum Density of PTFE/Cu can be obtained:(4)$$TMD=\frac{\rho_{PTFE}\cdot \rho_{Cu}}{C_{PTFE}\cdot \rho_{Cu}+C_{Cu}\cdot \rho_{PTFE}}$$

The actual density is determined by using the buoyancy method of the Archimedes principle and MH-300G multifunctional density balance *ρ*. The meaning of relative density is TMD/ρ%.

To study the effect of density on the mechanical properties of PTFE/Cu, three types of PTFE/Cu materials were designed in different ratios. Table 1 summarizes the component mass fractions and densities of each group of materials.

### 2.2. Microstructure of Composites

The powder morphologies of PTFE and Cu are illustrated in Figure 1. The initial powders of PTFE and Cu were loaded together in a ball mill and mechanically ground at 500 rpm for 12 h to disperse Cu into PTFE. After being ground, the PTFE powder and Cu could not maintain their initial shapes. The PTFE powder collided with the Cu powder and grinding ball and deformed in the grinding chamber because of the high strength of Cu and the stainless-steel ball. In addition, stripes appeared on the surface of the deformed PTFE powder due to the shear slip caused by high-speed rotation in the grinding chamber. The morphology of the PTFE powder resembles that of a deformed soft metal powder during grinding. The PTFE powder deforms after it is flattened, torn, and stacked into other powders. During grinding, Cu is embedded in and firmly attached to the surface of the PTFE powder. Cu on PTFE is indicated by red arrows in Figure 1b,c. Cu disperses in the grinding process, in which the metal is bonded to the powder surface, and the powder attached to Cu repeatedly experiences the process of flat deformation, tearing and stacking into other powder attached to Cu; Cu hardly aggregates and partially peels off.

Surface micrographs (Phenom TM, Phenom-World BV, Eindhoven, The Netherlands) of the extruded samples at the 500- and 100-μm scales were observed. Evident furrows and holes were observed on the surface of the mixture, which was caused by the preparation process. Microcracks appeared during the bonding of PTFE and Cu particles because of incompact bonding. These microcracks may develop into macrocracks, affecting the mechanical properties of the PTFE/Cu composite. The 10-μm Cu particles were surrounded by the PTFE particles. The surface of PTFE was lamellar and formed a dimple in the middle, as observed from the side view. We rationalized that filling the holes of the dimple with Cu particles would enhance the mechanical properties of the composites.

### 2.3. SHPB Experiment

The primary components of the SHPB experiment were a launch system, a test system, as well as a data acquisition and processing system. The experimental device is displayed in Figure 2. The loading system was composed of an air compressor, an air chamber, an impact bar, an incident bar, a transmission bar, a buffer bar, an energy absorber and rod-positioning support-bearing frame, an air pipe, and a trigger switch. The data acquisition system consisted of a measuring instrument with two parallel laser tubes and a double-path time measuring instrument, a stress wave signal collecting and recording device with a resistance strain gauge, an ultra-dynamic resistance strain gauge, a wheatstone bridge, and a dynamic data acquisition instrument (Nanjing University of Science and Technology, Nanjing, China) in the middle of the pressure bar.

We chose a cylindrical shape for the samples, with dimensions expressed in φ D × H, where D is the diameter of the sample and H is the height. For the dynamic compression experiment of materials, D is approximately 80% of the rod diameter; therefore, the dimensions of the sample in this experiment were φ 8 mm × 4 mm. The length of the impaction rod was 150 mm, and a highly sensitive semiconductor strain gauge was used to detect the waveform signal. A high-speed camera (Photron, FASTCAM SA-X2, Tokyo, Japan) was used to record the deformation of the specimen under an impact rate, and the dynamic deformation and failure behavior of the material was analyzed. The axial deformation of the specimen was monitored, and the compression and cracking behaviors were observed at a frequency of 50 MHz every 20 μs.

Figure 3 reveals the deformation progression of PTFE/Cu after impact. At 20 μs, sample end compression deformation occurred; the impact at 40 μs caused sample expansion in the middle and continued to be compressed to 180 μs when the PTFE/Cu sample began to fracture, including radial directions. Figure 4 depicts the failed PTFE/Cu samples.

## 3. Results and Discussion

### 3.1. Effect of Cu Powder on Mechanical Properties of PTFE

The dynamic compression properties of the PTFE/Cu materials were measured at different initial air pressures. The incident and transmitted wave signals generated in the experiment were collected and processed using a dynamic strain gauge (Archimedes Industrial Technology Co., Ltd., Beijing, China). The gains of the incident and transmitted wave signals were set to 500 and 1000, respectively. The dynamic compression stress–strain curves of the PTFE/Cu materials at different strain rates can be calculated using the two-wave method to process the experimental signals. The mechanical properties of the PTFE/Cu materials with different densities were analyzed. Table 2 presents the dynamic compression test data of PTFE/Cu under different strain rates for comparison. Both yield and compressive strengths increased with the strain rate, demonstrating the strain rate strengthening phenomenon of PTFE/Cu, and consequently its sensitivity to strain rate.

The determination of the stress–strain behavior of the material being tested in a Hopkinson bar is based on the same principles of one-dimensional elastic wave propagation within the pressure loading bars.

We use subscripts 1 and 2 to denote the incident and transmitted sides of the sample, respectively. Thereafter, we designate the strain in the bars as $\varepsilon_{i}$, $\varepsilon_{r}$, and $\varepsilon_{t}$, and the displacement at the end of the samples as *U*_1_ and *U*_2_ (input bar–sample and sample–output bar interfaces, respectively), as shown schematically in the magnified view of the sample in Figure 5.

From the linear superposition principle of elastic waves, the displacement at interfaces 1 and 2 can be written as follows:(5)$$U_{1}=c_{b}\int_{0}^{t}\varepsilon_{i}-\varepsilon_{r}d\tau$$
(6)$$U_{2}=c_{b}\int_{0}^{t}\varepsilon_{t}d\tau$$
where *c_b_* is the wave speed in the rod.

By definition, the average strain in the sample is expressed as:(7)$$\varepsilon \left(t\right)=\frac{U_{1}-U_{2}}{l_{s}}=\frac{c_{b}}{l_{s}}\int_{0}^{t}(\varepsilon_{i}-\varepsilon_{r}-\varepsilon_{t})d\tau$$

Differentiating Equation (7) with respect to time, the strain rate in the sample becomes:(8)$$\dot{\varepsilon }=\frac{c_{b}}{l_{s}}\left(\varepsilon_{i}-\varepsilon_{r}-\varepsilon_{t}\right)$$

By definition, the forces in the two bars are:(9)$$F_{1}=AE\left(\varepsilon_{i}+\varepsilon_{r}\right)$$
(10)$$F_{2}=AE\varepsilon_{t}$$
where *A* is the cross-sectional area of the pressure bar, and *E* is the Young’s modulus of the bars (considered equal, as the input and output bars are made of identical materials). From the one-dimensional elastic wave theory, we know that the sample is in force equilibrium. Therefore, by assuming that the sample deforms uniformly, we can equate the forces on each side of the sample, that is, *F*_1_ = *F*_2_. Comparing Equations (9) and (10).
(11)$$\varepsilon_{i}+\varepsilon_{r}=\varepsilon_{t}.$$

Substituting this criterion into Equations (7) and (8) yields:(12)$$\varepsilon \left(t\right)=2\frac{c_{b}}{l_{s}}\varepsilon_{r}d\tau$$
(13)$$\dot{\varepsilon }=\frac{2c_{b}}{l_{s}}\varepsilon_{r}$$

The stress was calculated from the strain gauge signal measure of the transmitted force divided by the instantaneous cross-sectional area (*A_s_*) of the sample:(14)$$\sigma \left(t\right)=\frac{AE\varepsilon_{t}}{A_{s}}$$
where σ(t) and ε(t) are functions describing the engineering stress and strain of the material, respectively (assuming that the material is incompressible). The relationship between the true stress and strain is expressed as:(15)$$\sigma_{T}=\left(1-\varepsilon \left(t\right)\right)\sigma \left(t\right)$$
(16)$$\varepsilon_{T}=-\ln(1-\varepsilon \left(t\right))$$

The density of the extruded PTFE/Cu sample affects its mechanical properties under dynamic compression. In addition, different densities have different strain rate effects. The mechanical response of the extruded samples with three densities under dynamic compression undergoes four stages, linear elastic, plastic yield, strain strengthening, and deformation failure. According to Figure 6, when the theoretical density of the sample was 2.5 g/cm^3^, the trends of the real stress–strain curves at the five strain rates were highly consistent. Table 2 and Figure 7 show that similar strain rate values yield similar mechanical property values, further demonstrating the uniformity and consistency of the extruded samples. At a strain rate of 5912 s^−1^, the maximum yield and compressive strengths were 50 and 79 MPa, respectively. In addition, the failure strain of samples increased with an increase in the compressive strength, albeit by a small margin. This result is attributed to the small difference in compressive strength.

For the sample with a theoretical density of 3.0 g/cm^3^, the mechanical properties were enhanced as this density increased. In addition, the strain rate effect of the sample is evident. The compressive strength, yield strength, and failure strain increased with only a subtle increase in the strain rate. When the strain rate reached a maximum value of 5447 s^−1^, the maximum yield strength, compressive strength, and failure strain were 52 MPa, 94 MPa, and 0.306 MPa, respectively.

At a theoretical density of 3.5 g/cm^3^, the mechanical properties were further enhanced as the density was increased. The strain rate effect of the sample became more evident. Its compressive strength increased with the strain rate, and consequently, both yield strength and failure strain increased. When the strain rate reached a maximum of 6.266 s^−1^, the maximum yield strength, compressive strength, and failure strain of the sample were 55 MPa, 110 MPa, and 0.337 MPa, respectively.

According to the stress–stress curves of the three materials at different strain rates, the three materials had varying degrees of the strain rate strengthening effect. To analyze the effect of material density on the dynamic mechanical properties of PTFE/Cu, the experimental data of the PTFE/Cu material with a true strain rate of approximately 4500 s^−1^ were selected for comparison through experiments.

Figure 8 depicts the relationship between material density and PTFE/Cu yield strength *σ_s_* and compressive strength *σ_bc_* at a strain rate of 4500 s^−1^. The yield and compressive strengths of the PTFE/Cu materials increased as the material density increased. This indicates that the filling modification of PTFE with Cu powder effectively improved the material’s resistance to compression under impact load, which will effectively improve the penetration performance of the PTFE expansive jet.

### 3.2. Microanalysis of Cu-Reinforced PTFE

The microstructure of the extruded PTFE/Cu composite was determined by backscattering SEM (Figure 9). These images reveal the formation of a continuous PTFE matrix with discretely distributed Cu particles. Several Cu particles were embedded in the PTFE matrix. In PTFE/Cu, the PTFE matrix plays the role of a connecting reinforcement phase and stress transfer. Cu particles are dispersed in the matrix as a reinforcing phase, and a considerable part of them are combined with the matrix as aggregates, which can prevent the occurrence of plastic deformation in the composite.

The Cu particle size of PTFE/Cu with different densities at a 50 μm scale was obtained by quantitative analysis of SEM images in Figure 9d–f. The histogram of particle size distribution in Figure 10 is obtained by analyzing the data. It can be seen from Figure 10 that when the Cu content is small, the distribution of Cu particles is relatively poor, and the increase of Cu particle content promotes the distribution of copper particles.

The microscopic images of the PTFE/Cu composite after impact deformation are illustrated in Figure 11. Numerous short and irregular cracks were observed, and the overall directivity was not evident. This reflects the continuous hindrance to the impact crack as it propagated in the direction of the tear of the specimen. The direction of the crack changes due to shortening, which raises the difficulty of crack propagation. Therefore, the impact failure energy required to break the sample becomes larger. Therefore, the impact resistance of the material can be improved by adding a certain amount of Cu particle filler. In addition to the factors of particles blocking the crack propagation, the interface bonding between Cu particles and PTFE (filament winding structure) will also increase the complexity of impact section formation, the surface area of the section and roughness.

During the formation of the fracture surface, the interfacial bonding had an extruding effect (interfacial debonding) on the particles at a certain depth below the fracture surface. Adding granular Cu (with a hardness remarkably higher than that of PTFE, and deformable) to PTFE can enable the composite to preferentially carry the load. Macromolecular chains can be adsorbed on the particle surface. Observe the state of Cu particles on the fracture surface according to Figure 11d–f; the Cu particle is wrapped by PTFE. Chains adsorbed by the Cu particles may also become entangled. Thus, when the PTFE/Cu composite is impacted, the stressed macromolecular chain can distribute the stress to each molecular chain through the particles, uniformly distributing the stress load and increasing the stress-causing microcracks in the matrix and the energy dissipated by the sample in the impact process. Thus, the composite becomes tougher.

According to the crack arrest theory, when a material is subjected to stress, the filling particles can cause the deflection and separation of the matrix microcracks (Ag striations), leading to the increase in fracture energy of the material. In addition, the interfacial layer may also inhibit the growth of the Ag grain; therefore, most of the kinetic and strain potential energies concentrated on the tip of Ag grains are transformed into discontinuous boundary movement properties. Therefore, the toughness (impact strength) of the PTFE/Cu composites is enhanced with 18–50.5% Cu.

### 3.3. Effect of Cu Powder on PTFE Jet Performance

The density of the shaped-charge liner has a significant influence on the jet-forming morphology and velocity distribution. Therefore, it is an important parameter in the design of the liner. The jet formation process with three different densities of the PTFE/Cu liner designed in this study is illustrated in Figure 10. The Johnson–Cook strength model and Shock state equation were used to describe the three types of liners. The parameters are listed in Table 3 [15]. Among them, the acoustic velocity of the PTFE/Cu liners was measured by the ultrasonic pulse reflection method and the numerical simulation method in our previous studies [13,14]. The main charge was explosive B, and the model was from the Autodyn-3D material library. The parameters are summarized in Table 4 [16].

As observed in Figure 12, the composite liner forms a stable expansive jet in the range of 2.46–3.38 g/cm^3^. According to the jet performance parameters shown in Table 5, the diameter and length of the expansive jet decreased, the dispersion decreased, and the cohesion increased as the density increased, because of the high acoustic velocity of Cu (3958 m/s). When the Cu content in the liner was increased, the acoustic velocity of the composite material also increased. In addition, as the mass of the charge increased, the collapse velocity of the charge wall element driven by the explosive decreased, and the jet velocity at the point of collision decreased. Thus, the gap between the acoustic velocity of the material and the jet velocity decreased. Further, the shockwaves generated by the charge wall element at the collision point weakened, so the jet cohesion gradually increased.

To study the behavior of the PTFE/Cu expansive jet, its formation process was observed by pulse X-ray technology and the morphology was obtained. In the experiment, two 450 kV pulse X-ray machines (HP Co.) were used for combined shooting. The experimental principle is described further in Figure 13. The two pulsed X-ray tubes were arranged at a certain angle and the shaped warhead was arranged vertically on the intersectional axis of the two X-ray tubes to ensure that the shaped jet flowed through it. By controlling the output time of the pulsed X-ray machines, two photographs of the jet morphology at different times were obtained on the X-ray photographic negative [14].

Figure 14 compares the pulse X-ray images of the PTFE and PTFE/Cu liner with the simulation results. The limited experiment conditions allowed us to capture only the X-ray pulse experiment results of pure PTFE and PTFE/Cu-3. The PTFE/Cu liner transformed the warhead into an effectively formed penetrator. The image contrast of the penetrator increased with the increase in material density. In addition, the jet head showed varying degrees of expansion effect, which is consistent with the theoretical and numerical predictions.

In the experimental image of the 16-μs X-ray pulse, the head of the jet approximately reaches the double stand-offs. Meanwhile, the jet profile is relatively clear, and the expansion effect is not evident, which can help clearly distinguish the jet and the pestles. The entire jet has good continuity and symmetry and maintains a high coaxiality with the detonation axis. In the image of the 31-μs pulse, the jet still has good continuity, symmetry, and coaxiality, but its profile becomes blurry, especially the jet head, with almost no clear boundary. This is mainly caused by the radial non-condensation of the jet, and meanwhile, an expansive jet is formed. In addition, compared with the jet diameter in the 16-μs image, the jet diameter in the 31-μs image is significantly thickened, the pestle body is considerably increased, the overall jet expands, the volume increases, and the density decreases. This is because the PTFE jet underwent volatile chemical reactions, and PTFE depolymerizes to form gaseous C_2_F_4_. The comparison between pulsed X-ray photos and simulation results reveals high consistency and convergence between the two jet morphologies.

A steel target experiment was conducted to demonstrate the effectiveness of the expansive jet of the PTFE/Cu composites with different densities. PTFE, PTFE/Cu-1, PTFE/Cu-2, and PTFE/Cu-3 were selected as the charge covers of the shaped warhead. Preliminary results show that the liners of these four materials could form varying degrees of expansion jets under the action of an explosive load. The standoff in the experiment was set to three times the charge diameter, and the steel target was vertically penetrated at a fixed explosion height cylinder. The results of the penetration experiment are displayed in Figure 15.

Figure 12 illustrates the results of the expansive jet from charge covers of four materials penetrating the steel target. The pure PTFE expansive jet could only create a small hole with a diameter of 14 mm and a penetration depth of 16 mm in the steel target. With 18% Cu powder added to the PTFE liner to increase the density, the PTFE/Cu-1 expansive jet created an 18-mm-wide, 20-mm deep hole in the steel target. Thus, the depth was increased by 25% with the addition of 18% Cu. At 37% Cu, the PTFE/Cu-2 jet drilled a 20-mm-wide, 26-mm-deep hole. Thus, the depth was increased by 62.5% with 37% Cu. At 50.5% Cu, the PTFE/Cu-3 jet created a 19-mm-wide, 29-mm-deep hole. Thus, the depth was increased by 81.3%. The results of the data analysis led to the conclusion that the damage power of the expansive jet, and consequently the hole diameter, could be notably improved by adding Cu powder in PTFE. The penetration hole barely changes with the increase in Cu content, whereas the penetration depth increases, enabling tactile warheads to achieve their purpose. This indicates that increasing the Cu content affects the quasistatic penetration stage but not the pitting stage of the penetration process, greatly improving the damage power of the polymer expansive jet. Thus, we contribute to the expansion of the application of polymer materials in shaped-charge warheads.

## 4. Conclusions

This study explores the application of the PTFE/Cu composite material in shaped-charge warheads. We added Cu powder to improve the strength of the expansive jet of pure PTFE penetrating a target plate. Three types of PTFE/Cu composites with different densities (2.46, 2.90, and 3.38 g/cm^3^) were prepared. The effect of increasing the PTFE/Cu density on the damage performance of the expansive jet was studied by a dynamic mechanical properties experiment, microscopic analysis, numerical simulation, and a penetration experiment.

1. The yield and compressive strengths of the PTFE/Cu composites increase with material density. This indicates that the filling modification of PTFE with Cu powder can improve the material’s resistance to compression under impact load, which will effectively improve the penetration performance of the PTFE expansive jet.

2. The microscopic analysis shows that the impact crack is continuously hindered in the process of expanding along the tearing direction of the sample. Because the crack becomes shorter, its direction changes and its difficulty of propagation increases, and the impact failure energy required to break the sample becomes larger. Therefore, the impact resistance of the PTFE/Cu composites improves after adding a certain amount of Cu particle filler. The toughness (impact strength) of the composites improves when 18–50.5% Cu powder is added.

3. The simulation and X-ray pulse experiment results demonstrate the formation of expansive jets from the PTFE/Cu composites with different densities. However, the diameter, length, and velocity of the PTFE/Cu expansive jets decrease linearly as the density increases. The experimental results show that the jet gains more damage capacity as the composite becomes denser. The penetration depth of the PTFE/Cu-1 (18% Cu) expansive jet in the steel target increases by 25%. The penetration depths of the PTFE/Cu-2 (37% Cu) and PTFE/Cu-3 (50.5% Cu powder) expansive jets increase by 62.5% and 81.3%, respectively.

## Figures and Tables

**Figure 1 polymers-14-02068-f001:**
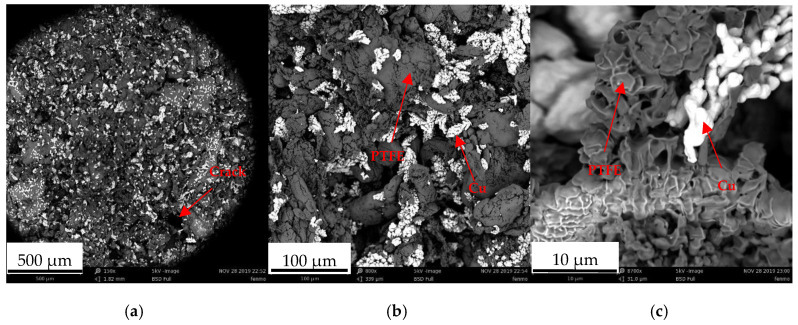
Scanning electron microscopy (SEM) micrograph of PTFE/Cu powder: (**a**) 500-μm scale; (**b**) 100-μm scale; (**c**) 10-μm scale.

**Figure 2 polymers-14-02068-f002:**
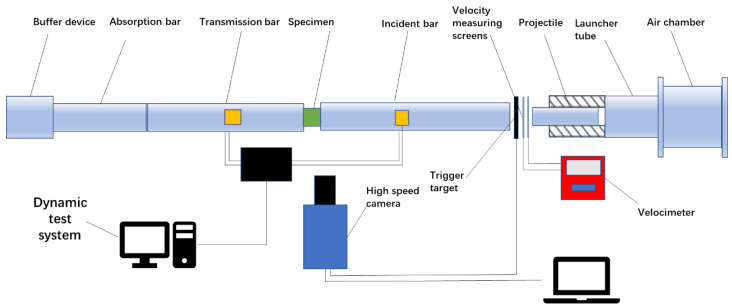
Schematic diagram of split-Hopkinson bar experimental device.

**Figure 3 polymers-14-02068-f003:**
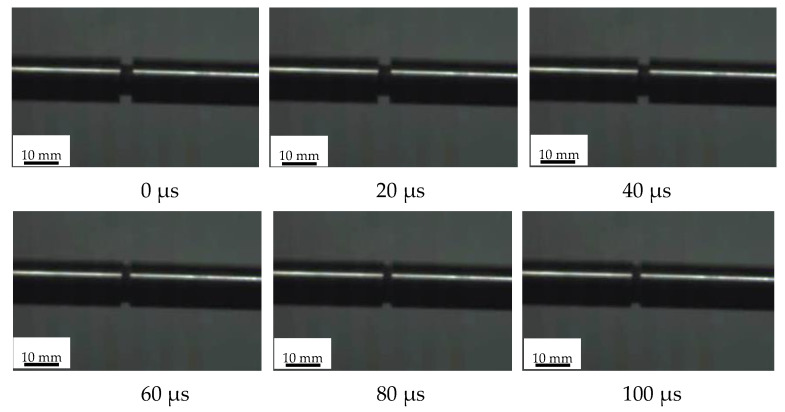
Instantaneous deformation process of PTFE/Cu at different impact times with impact speed of 20.8 m/s.

**Figure 4 polymers-14-02068-f004:**
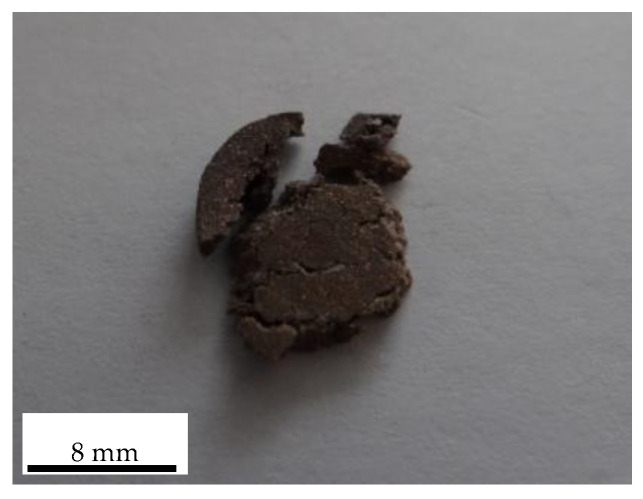
Failed PTFE/Cu sample.

**Figure 5 polymers-14-02068-f005:**
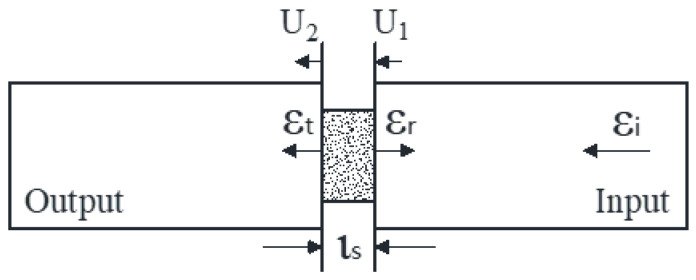
Expanded views of input bar/sample/output bar region.

**Figure 6 polymers-14-02068-f006:**
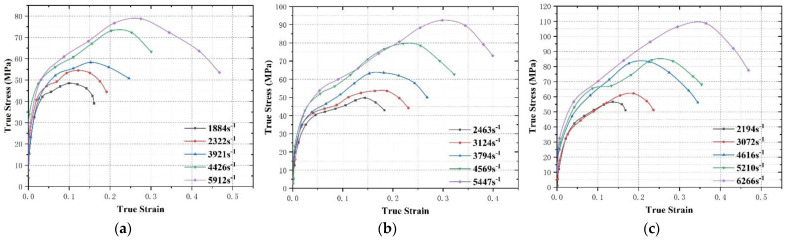
True stress–strain curves of PTFE/Cu with different densities under different strain rates: (**a**) PTFE/Cu-1; (**b**) PTFE/Cu-2; (**c**) PTFE/Cu-3.

**Figure 7 polymers-14-02068-f007:**
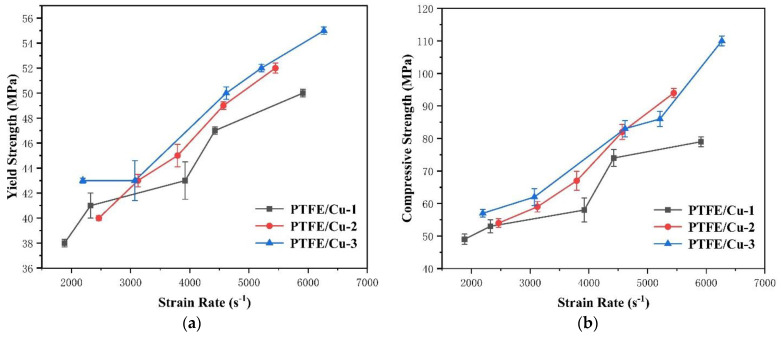
Parameters of dynamic mechanical properties of PTFE/Cu composites with different densities: (**a**) yield strength; (**b**) Compressive strength.

**Figure 8 polymers-14-02068-f008:**
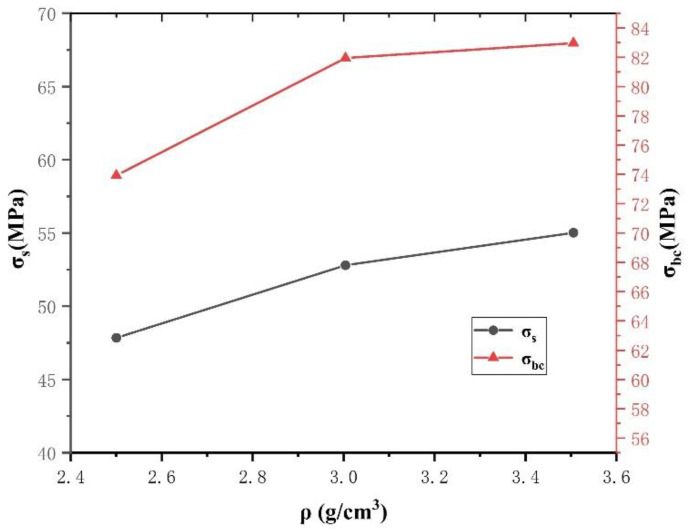
Effect of density on dynamic compression properties of PTFE/Cu.

**Figure 9 polymers-14-02068-f009:**
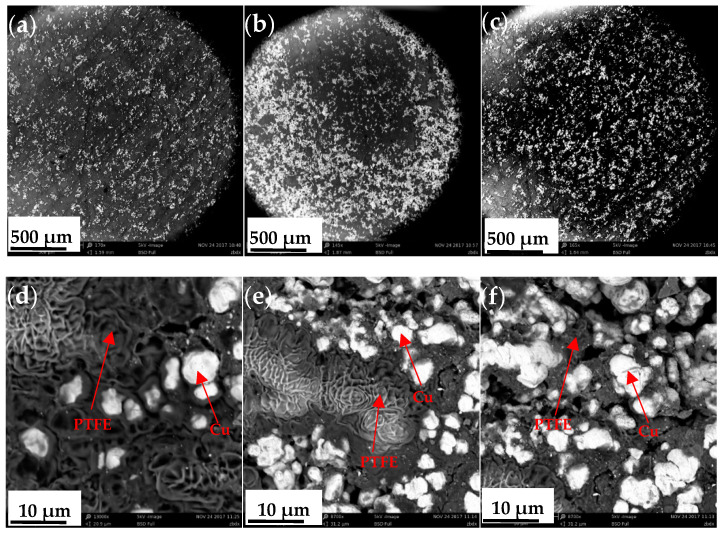
Backscattered SEM images of PTFE/Cu composites: (**a**) 500-μm scale PTFE/Cu-1; (**b**) 500-μm scale PTFE/Cu-2; (**c**) 500-μm scale PTFE/Cu-3; (**d**) 10-μm scale PTFE/Cu-1; (**e**) 10-μm scale PTFE/Cu-2; (**f**) 10-μm scale PTFE/Cu-3.

**Figure 10 polymers-14-02068-f010:**
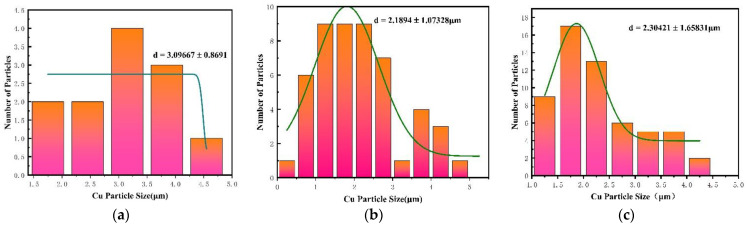
SEM image statistics of Cu particle size in PTFE/Cu materials with different density at 10μm scale: (**a**) PTFE/Cu-1; (**b**) PTFE/Cu-2; (**c**) PTFE/Cu-3.

**Figure 11 polymers-14-02068-f011:**
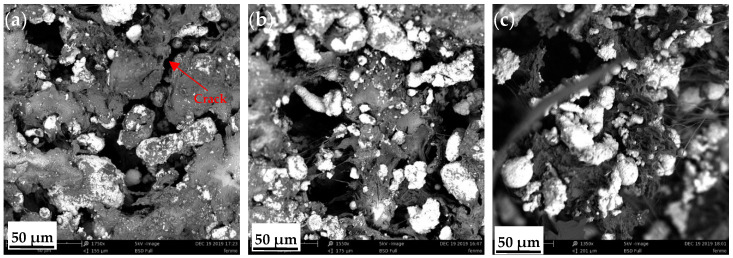
SEM images of impact sections in PTFE/Cu composites with different densities: (**a**) 50-μm scale PTFE/Cu-1; (**b**) 50-μm scale PTFE/Cu-2; (**c**) 50-μm scale PTFE/Cu-3; (**d**) 10-μm scale PTFE/Cu-1; (**e**)10-μm scale PTFE/Cu-2; (**f**) 5-μm scale PTFE/Cu-3.

**Figure 12 polymers-14-02068-f012:**
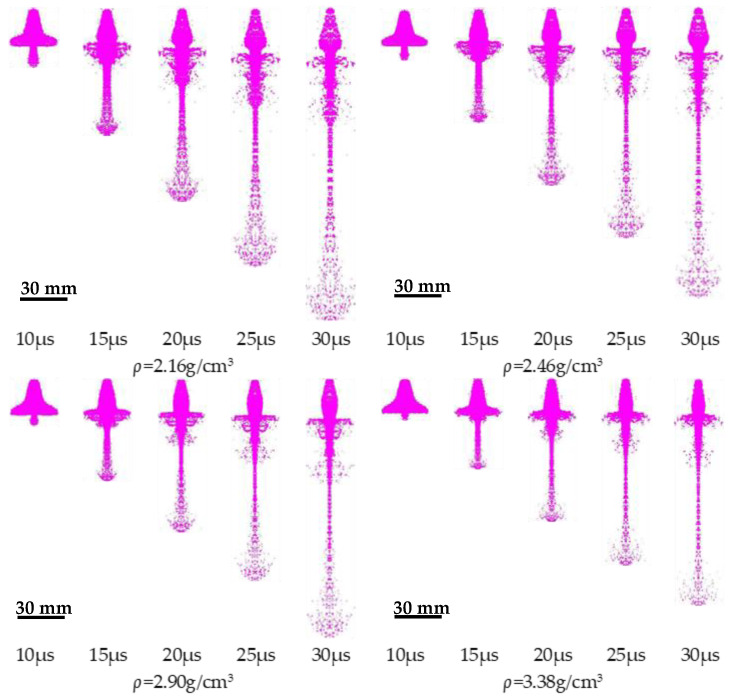
Jet forming process of liner with different densities.

**Figure 13 polymers-14-02068-f013:**
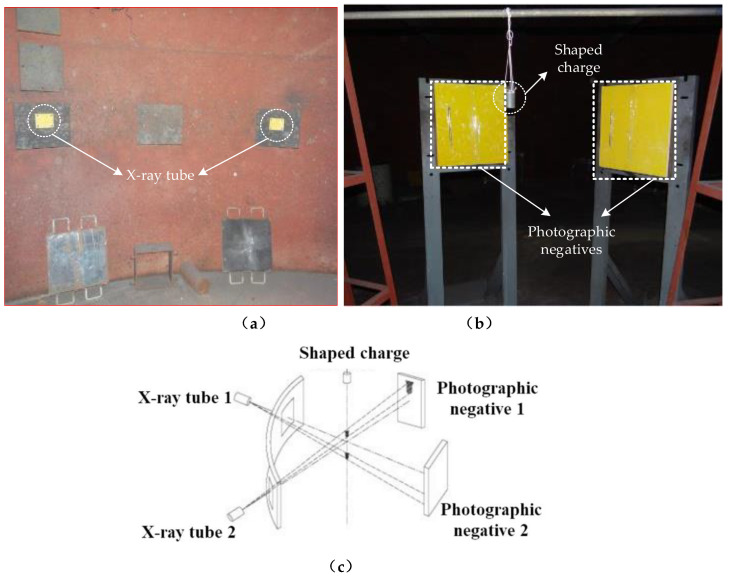
Field layout of pulse X-ray test: (**a**) Test scene graph; (**b**) Test layout; (**c**) Pulse X-ray test principle.

**Figure 14 polymers-14-02068-f014:**
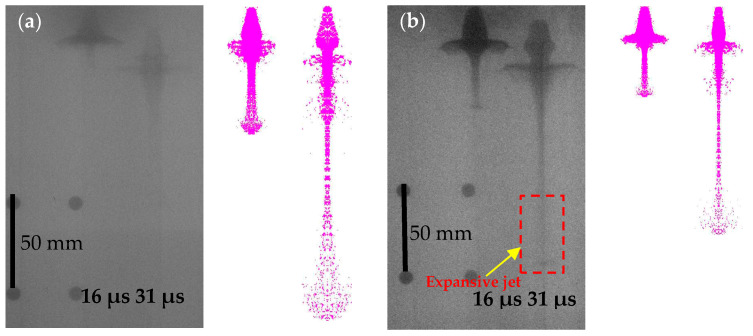
Comparison of X-ray images and numerical simulation results of expansive jet: (**a**) PTFE expansive jet; (**b**) PTFE/Cu-3 expansive jet.

**Figure 15 polymers-14-02068-f015:**
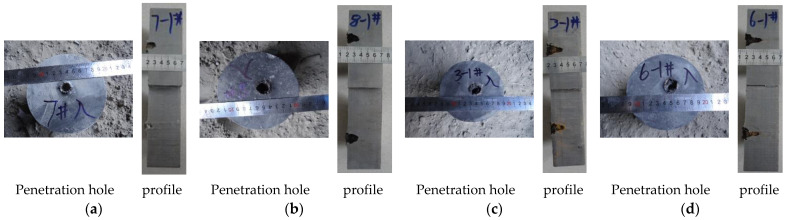
Results of target penetration by expansive jet from warheads of different materials: (**a**) PTFE; (**b**) PTFE/Cu-1; (**c**) PTFE/Cu-2; (**d**) PTFE/Cu-3. (The accuracy of the scale is 0.5 mm).

**Table 1 polymers-14-02068-t001:** Mass fraction and density of PTFE/Cu: PTFE: polytetrafluoroethylene; TMD: theoretical maximum density.

No.	Mass Fraction		TMD (g/cm^3^)	Density (g/cm^3^)	Relative Density (%)
	PTFE (%)	Cu (%)			
PTFE/Cu-1	82	18	2.50	2.46	98.40
PTFE/Cu-2	63	37	3.00	2.90	96.67
PTFE/Cu-3	49.5	50.5	3.50	3.38	96.57

**Table 2 polymers-14-02068-t002:** Mechanical parameters of PTFE/Cu composites at different strain rates.

Sample No.	Strain Rate (s^−1^)	Failure Strain	Yield Strength (MPa)	Compressive Strength (MPa)
PTFE/Cu-1	1884	0.101	38	49
	2322	0.116	41	53
	3921	0.157	43	58
	4426	0.223	47	74
	5912	0.252	50	79
PTFE/Cu-2	2463	0.142	40	54
	3124	0.175	43	59
	3794	0.193	45	67
	4569	0.231	49	82
	5447	0.306	52	94
PTFE/Cu-3	2194	0.136	43	57
	3072	0.176	43	62
	4616	0.205	50	83
	5210	0.253	52	86
	6266	0.337	55	110

**Table 3 polymers-14-02068-t003:** Main material parameters of liners.

Materials	*ρ* (g/cm^3^)	*A* (MPa)	*B* (MPa)	*n*	*C*	*C*_0_ (m/s)	*S*	*γ*
PTFE	2.16	8.04	75.94	1.01	0.16	1350	1.48	0.75
PTFE/Cu-1	2.46	10	29.94	1.02	0.18	1400	1.41	0.99
PTFE/Cu-2	2.90	10.5	32.58	1.12	0.22	1580	1.36	1.15
PTFE/Cu-3	3.38	11.4	33.28	0.97	0.25	1675	1.31	1.30

**Table 4 polymers-14-02068-t004:** Material parameters of B explosive.

*A* (Mbar)	*B* (Mbar)	R_1_	R_2_	ω	*D*_C-J_ (m/s)	*E* (GJ/m^3^)	*P*_C-J_ (GPa)
5.242	0.0768	4.2	1.1	0.34	7980	8.5	29.5

**Table 5 polymers-14-02068-t005:** Performance parameters of PTFE/Cu jet with different densities.

Liner Materials	Jet Velocity (m/s)	Jet Length (mm)	Jet Diameter (mm)
PTFE	6899	294.4	30
PTFE/Cu-1	6496	271.4	29
PTFE/Cu-2	5614	243	28
PTFE/Cu-3	5438	224.3	27

## Data Availability

Not applicable.

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
