# Peer review of "Experimental Study on Damage Characteristics of Copper-Reinforced Polytetrafluoroethylene Shaped-Charge Warhead Liner"

_polymers, 2022, doi:10.3390/polym14102068_

Round 1

Reviewer 1 Report

The article deals with the properties of PTFE / Cu composites depending on the proportion of Cu in the designed composites and the application of composites in shaped-charge warheads.

I have the following comments on the article:

- line 41 references [16]. The above work [16] does not agree with the references given on page 12-13.

- Table 1- all abbreviations must be explained before they are used in the article for the first time. The abbreviation TMD given in Table 1 is not given anywhere and explained.

- Fig. 1 What specific information does this image provide? In my opinion, the picture is redundant.

- What equipment was used for the preparation of cylindrical test specimens (Type, manufacturer ..)

- Specify the type and manufacturer of the Ball mill used

- What type of High Speed Camera was used for the experiment?

Author Response

Dear Editor, Reviewer

Thanks for your letter and the reviewers’ comments concerning our manuscript entitled “Experimental Study on Damage Characteristics of Copper-reinforced Polytetrafluoroethylene Shaped-charge Warhead Liner” (Manuscript ID: polymers-1708892). Those comments are all valuable and very helpful for improving our paper, as well as the important guiding significance to our researches. We have studied comments carefully and have made significant changes in the manuscript.

Response to reviewer’s comments

Reviewer:

Point 1: line 41 references [16]. The above work [16] does not agree with the references given on page 12-13.

Response 1: Thank you for underlining this deficiency. We have modified this error by changing reference [16] to reference [22]. The revised content is marked in red font in the manuscript.

Point 2: Table 1- all abbreviations must be explained before they are used in the article for the first time. The abbreviation TMD given in Table 1 is not given anywhere and explained.

Response 2: Thank you for underlining this deficiency. Theoretical Maximum Density (TMD), It was modified in the original manuscript and marked with red font.

Point 3: Fig. 1 What specific information does this image provide? In my opinion, the picture is redundant.

Response 3: Thank you for the suggestion. According to your opinion, after considering the integrity of the manuscript, we decided to delete Figure 1.

Point 4: What equipment was used for the preparation of cylindrical test specimens (Type, manufacturer ..)

Response 4: We are extremely grateful to reviewer for pointing out this problem. The extruder adopts PFB150 (Shang Ke, China), the working temperature is 260 ℃, and the screw speed is 20 ~ 30r / min. It was modified in the original manuscript and marked with red font.

Point 5: Specify the type and manufacturer of the Ball mill used

Response 5: We are extremely grateful to reviewer for pointing out this problem. The ball mill adopts planetary mill machine SN4.0 (Chao Yue, China), with a maximum volume of 4L and operates at room temperature. It was modified in the original manuscript and marked with red font.

Point 6: What type of High Speed Camera was used for the experiment?

Response 6: Thank you for your precious comments and advice. The High-speed camera used in the experimental is Photron (FASTCAM SA-X2, Japan). It was modified in the original manuscript and marked with red font.

We have tried our best to improve the manuscript and made significant changes in the manuscript. We appreciate Editors/Reviewers’ warm work earnestly, and hope that the correction will meet with approval. Once again, thank you very much you’re your comments. If there are any problems, please do not hesitate to contact with us.

Reviewer 2 Report

The paper entitled "Experimental Study on Damage Characteristics of Copper-reinforced Polytetrafluoroethylene Shaped-charge Warhead Liner" from Jianya Yi et al. focuses on the development and testing of PTFE-Cu composites. Dynamic mechanical properties are targeted. 

Main concerns:

1) Introduction section: The mechanical properties of powder polymer / particle reinforcing agents depend on the dispersion efficiency and interface compatibilization between the reinforcing agent and the matrix. There is no information about optimizing dispersion and interface bonding for this material system. It is pity that the authors did not try to optimize those two aspects based on the existing knowledge in the state of the art.

2) Introduction section: the previous paper of the authors (https://doi.org/10.1155/2021/5518172 in Advances in Materials Science and Engineering (2021, Article ID 5518172) is not described in the introduction. As a conclusion of this previous reference, it is indicated by the authors that sample preparation by hot-press sintering provides better mechanical properties than the sample preparation by extrusion. The current paper present results from extruded materials, so what is the point in continuing with extrusion? Can the authors explain the added value of the current study.

3) Experimental section: there is an important lack of details concerning material preparation and testing. For example, what is the exact copper reference and initial average size? What is the initial average size of PTFE powder ? Was it surface treated? Can the author describe the material mixing (ball milling machine reference, material volume, temperature...) and extrusion (machine reference, speed, temperature....) in details? How were determined density and relative density? How was extracted the stress-strain curves of SHPB Experiment?....  This section has to be significantly improved.

4) There is no error bars / standard deviation concerning mechanical properties

5) Can quantitative data be extracted from SEM images? For example, it could be relevant to determine the average particle size of Cu. This would enable to estimate the dispersion state based on the average particle size and comparing it with the average particle size prior to processing (dispersion state expected to be low). The Figure 8b exhibits a lack of distribution (meaning that the density of particle is not homogeneous), can a quantification of distribution be done based on SEM imaging?

6) The discussion on the presence of "Multiple macromolecular chains can be adsorbed on the surface of one Cu particle" should be based on a experimental evidence, there is currently no data proving a good interface bonding between Cu and PTFE. The simplest method would be to study fractured surface and see if some PTFE residues remain on the copper particle. If no interaction could be proved this sentence (and the similar ones) can be removed.

7) the authors indicate that "Cu particles, as a reinforcement phase, disperse evenly in the matrix and form a relatively complete island structure with the matrix" is not correct, indeed the Cu particles did not disperse since only agglomerates are visible, unless the authors can show single particle dispersion by TEM. In all the document, there is only a distribution of Cu agglomerates within the PTFE matrix (this distribution being not optimal in some cases as in Figure 8b.

Minor concerns:

1) Please define all the acronyms (for example, what is TMD in Table 1? and what is PDA in the introduction?)

2) Please provide a scale or unit in Figure 1 (unit?), Figure 4 (scale and unit), Figure 5 (scale and unit), Figure 100 and 11 (scale and unit), and Figure 12 (unit).

3) What is the origin of the material parameters provided in Tables 3 and 4? How were obtained the x-ray images in Figure 11? 

4) The experimental data with two digits are senseless, please only keep one digit.

Author Response

Dear Editor, Reviewer

Thanks for your letter and the reviewers’ comments concerning our manuscript entitled “Experimental Study on Damage Characteristics of Copper-reinforced Polytetrafluoroethylene Shaped-charge Warhead Liner” (Manuscript ID: polymers-1708892). Those comments are all valuable and very helpful for improving our paper, as well as the important guiding significance to our researches. We have studied comments carefully and have made significant changes in the manuscript.

Response to reviewer’s comments

Reviewer:

Main concerns:

Point 1: Introduction section: The mechanical properties of powder polymer / particle reinforcing agents depend on the dispersion efficiency and interface compatibilization between the reinforcing agent and the matrix. There is no information about optimizing dispersion and interface bonding for this material system. It is pity that the authors did not try to optimize those two aspects based on the existing knowledge in the state of the art.

Response 1: We are grateful for the suggestion. Regarding the particle dispersion and interface bonding of composite materials, relevant references have been added. In our previous research, we considered improving the particle dispersion by ball milling, but the particle distribution of the sample prepared by extrusion process is still not ideal. In the future, we will make an in-depth analysis of these two problems according to your suggestion. The revised part is on lines 43 and 68, from page 1 to page 2. The revised part is as follows:

Regarding the particle dispersion and interface bonding of composite materials, Wang [23] studied the successful preparation of aluminum matrix composites with high bo-ron nitride nanosheets (BNNSs) content by variable speed ball milling, gradual addition of BNNSs and finally direct current sintering (DCS). Such ball milling method can effectively disperse BNNSs onto the existing Al surface and the fresh Al surface generated by ball milling, allowing for a high content of BNNSs homogeneously dispersed within the composites. Danaya [24] found that the morphology of polybutylene-adipate-co-terephthalate (PBAT) / thermoplastic star (TPS) showed aggregation of nanoparticles, resulting in poor mechanical properties. The interaction between ZnO nano filler and polymer increases the dispersion of nanoparticles and reduces the ag-glomeration of nanoparticles. Zhao [25] prepared reduced graphene oxide (RGO) / Cu Matrix Composites by electrostatic adsorption method with interface transition phase design. Adding Cu / Ti alloy powder can improve the bonding by forming carbides at the RGO / Cu interface, and finally improve the mechanical properties of the compo-sites. Li [26] studied the preparation of composite solders with different graphene dispersion by different ball milling methods. Microstructure characterization showed that incoherent and amorphous interfaces were formed between graphene and tin. Guo [27] obtained a non-equilibrium interface that can provide tight interfacial bonding be-tween the carbon nanotubes (CNTs) and Al matrix in the Al/CNTs composites fabricated through spark plasma sintering (SPS) and subsequently hot extrusion. This special interface, accompanied by small grain size, the uniform dispersion, and the integrity of the CNTs, can significantly improve the mechanical properties of the Al/CNTs composite. In our last study [28], the mechanical properties of PTFE / Cu samples pre-pared by hot pressing sintering process and extrusion process were discussed. The results show that the mechanical properties of the samples prepared by hot pressing sintering process are better than those prepared by extrusion process, but the penetration performance is weak as a liner.

Point 2: Introduction section: the previous paper of the authors (https://doi.org/10.1155/2021/5518172 in Advances in Materials Science and Engineering (2021, Article ID 5518172) is not described in the introduction. As a conclusion of this previous reference, it is indicated by the authors that sample preparation by hot-press sintering provides better mechanical properties than the sample preparation by extrusion. The current paper present results from extruded materials, so what is the point in continuing with extrusion? Can the authors explain the added value of the current study.

Response 2: Thank you for your careful review. A discussion about this article is added in the introduction. The revised part is on lines 64 and 68. On the second page of the revised manuscript, the revised part is marked in red font. The following is an explanation of the added value of the current study.

In the last paper, PTFE / Cu composites prepared by different processes were studied. The results showed that the mechanical properties of PTFE / Cu composites prepared by hot pressing sintering were better than those prepared by extrusion. However, in practical application, it was found that PTFE / Cu composites prepared by extrusion process had two advantages in the application of shaped charge warfare.The first advantage is that after the PTFE / Cu composite bar is extruded, the liner with different structural parameters can be made through simple machining. When the liner with different mechanism parameters is prepared by hot pressing sintering process, different hot pressing sintering molds need to be designed, which is not conducive to the damage performance analysis of the combat part.

Secondly, through the early penetration experiment, it is found that the damage ability of the liner manufactured by hot pressing sintering process is weaker than that of extrusion process, and does not achieve the expected damage power. Therefore, this paper mainly studies the preparation of PTFE / Cu liner by extrusion process, and analyzes the influence of different density on the damage power of PTFE / Cu expansive jet. However, one of the difficulties encountered is that the extrusion process cannot further improve the content of Cu, which also requires us to improve the preparation process.

Point 3: Experimental section: there is an important lack of details concerning material preparation and testing. For example, what is the exact copper reference and initial average size? What is the initial average size of PTFE powder? Was it surface treated? Can the author describe the material mixing (ball milling machine reference, material volume, temperature...) and extrusion (machine reference, speed, temperature....) in details? How were determined density and relative density? How was extracted the stress-strain curves of SHPB Experiment?....  This section has to be significantly improved.

Response 3: Our deepest gratitude goes to you for your careful work and thoughtful suggestions that have helped improve this paper substantially. We revised the comments on the manuscript, and marked the revised part in red font.

a、what is the exact copper reference and initial average size?

The particle size of copper powder used in the sample and the liner in this test is 3~5 μm.

b、What is the initial average size of PTFE powder?

The average particle size of PTFE used in this experimental study is 220 μm, with an average density of 2.2g/cm3.

c、Was it surface treated?

The surface was not treated, but during the damage test, a simple grinding was carried out during the assembly process.

d、Can the author describe the material mixing (ball milling machine reference, material volume, temperature...) and extrusion (machine reference, speed, temperature....) in details?

The ball mill adopts planetary mill machine SN4.0 (Chao Yue, China), with a maximum volume of 4L and operates at room temperature.

The extruder adopts PFB150 (Shang Ke, China), the working temperature is 260 ℃, and the screw speed is 20 ~ 30r / min.

e、How were determined density and relative density?

Determine TMD(Theoretical Maximum Density):

If the mass fraction of PTFE and Cu powder is CPTFE and CCu respectively, and the volume is VPTFE and VCu respectively, there is

                            (1)

                              (2)

                           (3)

Substituting formula (1) and formula (2) into formula (3), it can be obtained that the Theoretical Maximum Density of PTFE / Cu is

                       (4)

Determine actual density ρ:

The actual density is determined by using the buoyancy method of Archimedes principle and MH-300G multifunctional density balance ρ.

Determine Relative Density:

Relative Density= %

f、How was extracted the stress-strain curves of SHPB Experiment?

The calculation method of stress-strain curve is added.

The determination of the stress–strain behaviour of the material being tested in a Hopkinson bar is based on the same principles of one-dimensional elastic wave propagation within the pressure loading bars.

We use subscripts 1 and 2 to denote the incident and transmitted sides of the sample, respectively. Thereafter, we designate the strain in the bars as , , and , and the displacement at the end of the samples as U1 and U2 (input bar–sample and sample–output bar interfaces, respectively), as shown schematically in the magnified view of the sample in Figure 5.

Figure 5. Expanded views of input bar/sample/output bar region

From the linear superposition principle of elastic waves, the displacement at interfaces 1 and 2 can be written as follows:

,                                                (5)

,                                            (6)

where  is the wave speed in the rod.

By definition, the average strain in the sample is expressed as

.                                     (7)

Differentiating Equation 7 with respect to time, the strain rate in the sample becomes

.                                                 (8)

By definition, the forces in the two bars are

,                                                      (9)

,                                                      (10)

where  is the cross-sectional area of the pressure bar, and  is the Young’s modulus of the bars (considered equal, as the input and output bars are made of identical materials). From the one-dimensional elastic wave theory, we know that the sample is in force equilibrium. Therefore, by assuming that the sample deforms uniformly, we can equate the forces on each side of the sample, that is, . Comparing Equations 9 and 10.

                                             (11)

Substituting this criterion into Equation 7 and 8 yields

,                                           (12)

.                                             (13)

The stress was calculated from the strain gauge signal measure of the transmitted force divided by the instantaneous cross-sectional area ( ) of the sample:

,                                                  (14)

where  and  are functions describing the engineering stress and strain of the material, respectively (assuming that the material is incompressible). The relationship between the true stress and strain is expressed as

,                                      (15)

.                                      (16)

Point 4: There is no error bars / standard deviation concerning mechanical properties

Response 4: Thank you for underlining this deficiency. The error analysis of mechanical property parameters is carried out, and the error bar of mechanical property parameters in Figure x is obtained. The revised part is marked in red font in the manuscript.

Figure 7. Parameters of dynamic mechanical properties of PTFE / Cu composites with different densities: (a) yield strength; (b) Compressive strength.

Point 5: Can quantitative data be extracted from SEM images? For example, it could be relevant to determine the average particle size of Cu. This would enable to estimate the dispersion state based on the average particle size and comparing it with the average particle size prior to processing (dispersion state expected to be low). The Figure 8b exhibits a lack of distribution (meaning that the density of particle is not homogeneous), can a quantification of distribution be done based on SEM imaging?

Response 5: We are very grateful to the reviewers for their suggestions. According to the reviewer's comments, we added a picture of SEM quantitative results. The Cu particle size of PTFE / Cu with different density at 50μm scale was obtained by quantitative analysis of SEM photos in Figure 8d, 8e and 8f. The histogram of particle size distribution in Figure 10 is obtained by analyzing the data. From Figure x, it can be obtained that increasing the content of Cu particles promotes the distribution of Cu particles. The revised part is marked in red font in the manuscript.

Figure 10. SEM image statistics of Cu particle size in PTFE / Cu materials with different density at 10μm scale: (a) PTFE/Cu-1; (b) PTFE/Cu-2; (a) PTFE/Cu-3.

Point 6: The discussion on the presence of "Multiple macromolecular chains can be adsorbed on the surface of one Cu particle" should be based on a experimental evidence, there is currently no data proving a good interface bonding between Cu and PTFE. The simplest method would be to study fractured surface and see if some PTFE residues remain on the copper particle. If no interaction could be proved this sentence (and the similar ones) can be removed.

Response 6: We apologize for the incorrect statement description the original manuscript. According to the guidance of review suggestions, the winding phenomenon between copper particles and PTFE shown in Figure 11 is obtained by observing the fracture surface of PTFE / Cu composite. Based on the observation, change the sentence to "Cu particle is wrapped by PTFE". We added SEM images of copper particles on the fracture surface on line 325 and page 11.

Figure 11. SEM images of impact sections in PTFE/Cu composites with different densities: (a) 50-μm scale PTFE/Cu-1; (b) 50-μm scale PTFE/Cu-2; (c) 50-μm scale PTFE/Cu-3; (d) 10-μm scale PTFE/Cu-1; (e)10-μm scale PTFE/Cu-2; (f) 5-μm scale PTFE/Cu-3.

Point 7: the authors indicate that "Cu particles, as a reinforcement phase, disperse evenly in the matrix and form a relatively complete island structure with the matrix" is not correct, indeed the Cu particles did not disperse since only agglomerates are visible, unless the authors can show single particle dispersion by TEM. In all the document, there is only a distribution of Cu agglomerates within the PTFE matrix (this distribution being not optimal in some cases as in Figure 8b.

Response 7: We are very sorry for the incorrect description of copper particle dispersion. I have changed " Cu particles, as a reinforcement phase, disperse evenly in the matrix and form a relatively complete island structure with the matrix " in the original manuscript to " Cu particles are dispersed in the matrix as reinforcing phase, and a considerable part of them are combined with the matrix as aggregates". We added SEM images of PTFE mixed Cu fracture surface in point 6, which shows some Cu particles and Cu agglomerates. Therefore, we need to further study the single particle dispersion. At present, due to the COVID-19, TEM verification cannot be carried out, but we will conduct a detailed study on the dispersion ability of copper particles in the future.

Minor concerns:

Point 8: Please define all the acronyms (for example, what is TMD in Table 1? and what is PDA in the introduction?).

Response 8: Thank you for underlining this deficiency. Theoretical Maximum Density(TMD)、polydopamine(PDA) , It was modified in the original manuscript and marked with red font.

Point 9: Please provide a scale or unit in Figure 1 (unit?), Figure 4 (scale and unit), Figure 5 (scale and unit), Figure 10 and 11 (scale and unit), and Figure 12 (unit).

Response 9: Thank you for the suggestion. We have revised all the Figures mentioned in the review comments and marked the scale and unit. Due to the non-necessity of Figure 1, we deleted Figure 1 on page 2. Figure 4 of the original manuscript is figure 3 in the revised manuscript, with the scale and unit marked. In the revised manuscript, lines 169 to 176 are on page 5. Figure 5 of the original manuscript is figure 4 in the revised manuscript, with the scale and unit marked. It is on line 184 and page 5 in the revised manuscript. Figures 10 and 11 of the original manuscript are figures 12 and 14 in the revised manuscript, with scale and units marked. Lines 350 and 382 in the revised manuscript are on page 12 and 13 respectively. Figure 12 of the original manuscript is Figure 15 in the revised manuscript, with the accuracy of the scale marked on line 405 and page 14 in the revised manuscript.

Point 10: What is the origin of the material parameters provided in Tables 3 and 4? How were obtained the x-ray images in Figure 11?

Response 10: We are extremely grateful to reviewer for pointing out this problem. The material parameters in tables 3 and 4 are from the AUTODYN software parameter library, and the references have been marked. X-ray images is obtained by X-ray pulse testing machine. How to obtain x-ray images is explained below. In the revised manuscript, lines 364 to 374, on pages 12 and 13.

To study the behavior of the PTFE/Cu expansive jet, its formation process was observed by pulse X-ray technology and the morphology was obtained. In the experiment, two 450 kV pulse X-ray machines (HP Co.) were used for combined shooting. The experimental principle is described further in Figure 13. The two pulsed X-ray tubes were arranged at a certain angle and the shaped warhead was arranged vertically on the intersectional axis of the two X-ray tubes to ensure that the shaped jet flowed through it. By controlling the output time of the pulsed X-ray machines, two photographs of the jet morphology at different times were obtained on the X-ray photographic negative [14].

Figure 13. Field layout of pulse X-ray test: (a) Test scene graph; (b) Test layout; (c) Pulse X-ray test principle.

Point 11: The experimental data with two digits are senseless, please only keep one digit.

Response 11: We deeply appreciate the reviewer’s suggestion. We have revised this opinion, and the revised part is marked in red font in the manuscript.

We have tried our best to improve the manuscript and made significant changes in the manuscript. We appreciate Editors/Reviewers’ warm work earnestly, and hope that the correction will meet with approval. Once again, thank you very much you’re your comments. If there are any problems, please do not hesitate to contact with us.

Round 2

Reviewer 2 Report

The paper has been significantly improved, much more details on the experimental section and on the results description and analysis have been provided. It can be now accepted in Polymers.